# Comorbidity Burden and Presence of Multiple Intracranial Lesions Are Associated with Adverse Events after Surgical Treatment of Patients with Brain Metastases

**DOI:** 10.3390/cancers12113209

**Published:** 2020-10-31

**Authors:** Matthias Schneider, Muriel Heimann, Christina Schaub, Lars Eichhorn, Anna-Laura Potthoff, Frank A. Giordano, Erdem Güresir, Yon-Dschun Ko, Jennifer Landsberg, Felix Lehmann, Alexander Radbruch, Katjana S. Schwab, Leonie Weinhold, Johannes Weller, Christian Wispel, Ulrich Herrlinger, Hartmut Vatter, Niklas Schäfer, Patrick Schuss

**Affiliations:** 1Department of Neurosurgery, Center of Integrated Oncology (CIO) Bonn, University Hospital Bonn, 53127 Bonn, Germany; muriel.heimann@ukbonn.de (M.H.); anna-laura.potthoff@ukbonn.de (A.-L.P.); erdem.gueresir@ukbonn.de (E.G.); christian.wispel@ukbonn.de (C.W.); hartmut.vatter@ukbonn.de (H.V.); patrick.schuss@ukbonn.de (P.S.); 2Division of Clinical Neuro-Oncology, Department of Neurology, Center of Integrated Oncology (CIO) Bonn, University Hospital Bonn, 53127 Bonn, Germany; christina.schaub@ukbonn.de (C.S.); johannes.weller@ukbonn.de (J.W.); ulrich.herrlinger@ukbonn.de (U.H.); niklas.schaefer@ukbonn.de (N.S.); 3Department of Anesthesiology and Intensive Care, University Hospital Bonn, 53127 Bonn, Germany; lars.eichhorn@ukbonn.de (L.E.); felix.lehmann@ukbonn.de (F.L.); 4Department of Radiation Oncology, Center of Integrated Oncology (CIO) Bonn, University Hospital Bonn, 53127 Bonn, Germany; frank.giordano@ukbonn.de; 5Department of Oncology and Hematology, Center of Integrated Oncology (CIO) Bonn, Johanniter Hospital Bonn, 53113 Bonn, Germany; Yon-Dschun.Ko@bn.johanniter-kliniken.de; 6Department of Dermatology and Allergy, Center of Integrated Oncology (CIO) Bonn, University Hospital Bonn, 53127 Bonn, Germany; jennifer.landsberg@ukbonn.de; 7Department of Neuroradiology, University Hospital Bonn, 53127 Bonn, Germany; alexander.radbruch@ukbonn.de; 8Department of Internal Medicine III, Center of Integrated Oncology (CIO) Bonn, University Hospital Bonn, 53127 Bonn, Germany; katjana.schwab@ukbonn.de; 9Institute of Medical Biometrics, Informatics, and Epidemiology, University Hospital Bonn, 53127 Bonn, Germany; leonie.weinhold@ukbonn.de

**Keywords:** brain metastases, surgical management, Charlson comorbidity index, cancer

## Abstract

**Simple Summary:**

Patients with brain metastasis must face advanced cancer diseases, and neurosurgical resection is often considered an inevitable part of treatment. However, peri- and postoperative complications might further worsen the prognosis for these vulnerable patients. It is therefore important to investigate risk factors for such unfavorable events in order to recognize high-risk patients at the earliest possible stage of disease. For this purpose, we aimed to identify risk factors for early postoperative complications following surgical resection of brain metastases. Our results showed that the presence of multiple brain metastases in a single patient and a high rate of additional comorbidities were associated with elevated levels of postoperative complications. Furthermore, patients who suffered from such unfavorable postoperative events were significantly more likely to die during the initial hospital stay. The present study therefore might help to preoperatively select for critically ill patients who are in mandatory need of advanced treatment and care.

**Abstract:**

Surgical resection is a key treatment modality for brain metastasis (BM). However, peri- and postoperative adverse events (PAEs) might be associated with a detrimental impact on postoperative outcome. We retrospectively analyzed our institutional database with regard to patient safety indicators (PSIs), hospital-acquired conditions (HACs) and specific cranial surgery-related complications (CSCs) as high-quality metric profiles for PAEs in patients who had undergone surgery for BM in our department between 2013 and 2018. The comorbidity burden was assessed by means of the Charlson comorbidity index (CCI). A multivariate analysis was performed to identify independent predictors for the development of PAEs after surgical resection of BM. In total, 33 patients (8.5%) suffered from PAEs after surgery for BM. Of those, 17 PSI, 5 HAC and 11 CSC events were identified. Multiple brain metastases (*p* = 0.02) and a higher comorbidity burden (CCI > 10; *p* = 0.003) were associated with PAEs. In-hospital mortality of patients suffering from a PAE was significantly higher than that of patients without a PAE (24% vs. 0.6%; *p* < 0.0001). Awareness of risk factors for postoperative complications enables future prevention and optimal response, particularly in vulnerable oncological patients. The present study identified the presence of multiple brain metastases and increased comorbidity burden associated with PAEs in patients suffering from BM.

## 1. Introduction

Patients with brain metastasis (BM) require multimodal therapy, which often includes surgical removal of the intracranial mass to meet the standard of care [1,2]. However, these neurosurgical procedures can also cause postoperative adverse events (PAEs), which can be a significant burden to the affected patients [3]. The occurrence of PAEs is clearly linked to a worse prognosis in the perioperative period. Therefore, it is important to identify potentially preventable PAEs and to recognize the corresponding risk factors. For this purpose, the Agency of Healthcare Research and Quality and the Center for Medicare and Medicaid Services have compiled a list of events known as patient safety indicators (PSIs) and hospital-acquired conditions (HACs). These are intended to serve as quality indicators that are used to evaluate the quality of medical care and are therefore often at the center of strategies to reduce complications. However, in the case of surgical procedures for patients with BM, these established quality indicators do not cover complications that are specific to cranial surgeries. Postoperative PAEs might be associated with an additional burden for the patient and the delay or interruption of the necessary adjuvant therapy that must be carried out in a timely manner. Consequently, it is mandatory to investigate potential risk factors for the development of postoperative complications following surgical treatment of patients requiring surgery for BM in order to ensure perioperative safety and quality of care for these patients.

## 2. Results

### 2.1. Patient Characteristics

Overall, 388 patients suffering from brain metastases were surgically treated between 2013 and 2018 at the authors’ institution. The median age was 64 years (range 20–91 years), including 191 female (49%) and 197 male (51%) patients. In total, 125 patients (32%) with surgically resected BM exhibited multiple intracranial lesions at the time of surgery. Patients with surgically treated BM suffered from a median age-adjusted Charlson comorbidity index (CCI) of 11 (range 8–16). The most common origin of the brain metastases examined was the lung (43%), followed by breast tumors (13%), gastrointestinal tumors (13%) and otherwise localized melanomas (11%). In 3% of patients with BM, the primary site of cancer remained unknown despite histopathological examination of the surgically removed BM and further staging investigations. Further details on the patient characteristics at the time of hospitalization are provided in Table 1.

### 2.2. Postoperative Complications

Overall, 33 patients (8.5%) suffered from an early PAE after surgical resection of BM. Subsequent analysis of early PAEs resulted in a subclassification into 17 PSIs (4.4%), 5 HACs (1.3%) and 11 specific cranial surgery-related complications (CSCs) (2.8%; Table 2). Postoperative hemorrhage and pneumonia were the most common PSIs and HACs, occurring in 11 (2.8%) and in 3 (0.8%) patients with BM, respectively. Additional PSIs identified consisted of three respiratory failures (0.8%) and two pulmonary embolisms (0.5%), as well as one wound dehiscence (0.3%). The subgroup of specific CSCs consisted of brain edema and postoperative meningitis in three patients each (1%), postoperative cerebrospinal fluid (CSF) leakage and postoperative seizures in two patients each (0.5%) and cerebrovascular venous thrombosis in one patient (0.3%).

ROC analysis of age-adjusted CCI in the present patient panel resulted in an area under the curve (AUC) of 0.63 (95% CI 0.54–0.72; *p* = 0.01). For further dichotomization of the age-adjusted CCI, a threshold value of >9.5 was derived, resulting in a sensitivity of 87.9% and a specificity of 73.8% for predicting the probability of a PAE. Patients with BM and a comorbidity burden in terms of a CCI > 10 suffered significantly more early postoperative adverse events compared to patients with a lower comorbidity burden (*p* = 0.003, OR 3.3, 95% CI 1.4–7.4). In addition, patients who developed a postoperative complication were significantly more likely to harbor multiple intracranial metastases when compared with patients without postoperative complications (*p* = 0.02, OR 2.4, 95% CI 1.2–4.9). The occurrence of postoperative complications also caused the patients affected to remain in the hospital for a significantly longer period of time when compared with patients without a PAE (26 ± 22 days vs. 14 ± 12 days; *p* < 0.0001, 95% CI 7.3–16.7). Furthermore, in-hospital mortality rate was significantly higher in patients suffering from a PAE when compared with patients without a PAE after surgical resection of BM (24% vs. 0.6%; *p* < 0.0001, OR 56.5, 95% CI 11.4–280.3).

Analysis of overall survival (OS) revealed that patients without a PAE exhibited a median OS rate of 11 months (95% CI 9–13), which was significantly greater (*p* < 0.0001) than the median OS rate of 2.5 months (95% CI 1–4 months) for patients with a PAE (Figure 1).

## 3. Discussion

According to the data of the present study, there was a significant increase in in-hospital mortality in patients with surgically treated brain metastases in cases of early postoperative adverse events. Both the presence of multiple BMs at the time of surgical treatment and a higher comorbidity burden seemed to result in a more likely development of postoperative complications.

Among the detrimental effects of cancer is the propagation of metastases into the brain with subsequent destructive effects on many essential functions crucial to the quality of daily life. Notably, brain metastases are, indeed, an indicator of poor prognosis in patients with solid cancers [4]. Nevertheless, neurosurgical tumor removal is one of the fundamental pillars of treatment for brain metastases. As the evidence for the indication of neurosurgical treatment of brain metastases is increasingly condensing into several established standards, the individual risk–benefit assessment of surgical intervention remains one of the most important tasks of the clinical (neuro)oncology team [5,6]. A major contribution to this individual patient assessment is the collaborative expert consultation across the boundaries of several medical disciplines, for example in the framework of interdisciplinary tumor board meetings [7]. Another important prerequisite in the treatment of patients with BM is the assessment of the peri- or postoperative risk profile of the applied treatment. The lower the adverse events of a particular treatment are, the better for the affected patients. A treatment that causes more harm than good is not desirable, especially for people with limited life expectancy. Adverse events, including mortality after surgery for metastatic brain tumors, were observed to be less frequent in high-volume centers [8]. These results are attributable not only to a more experienced surgical team, but also to the sharpened, more established standards, which have become necessary due to the higher volume of cases and which are only made possible by a trustworthy, interdisciplinary collaboration. In addition to the chance of increasing treatment safety given a high patient volume, it is also necessary to address potential risk factors for the occurrence of postoperative complications.

Apart from these considerations, it is necessary to analyze potential risk factors for the occurrence of disease-specific complications. In the present study, the preoperative presence of multiple intracranial metastatic lesions was observed as a potential influence on postoperative complications after brain metastasis surgery. Besides the well-established treatment strategy for patients with a single BM, decision-making for the optimal treatment of patients with multiple metastases is challenging [9]. Traditionally, patients with multiple BMs have been considered poor candidates for aggressive local treatments such as surgery [10]. However, in addition to the possibility of reducing the tumor burden, the surgical treatment of patients with multiple metastases also offers the possibility of “gaining space”. Thereby, a certain safety margin is offered to subsequent treatment methods, such as postoperative radiation, in order to compensate for the therapy-associated brain swelling and therefore reduce neurological worsening. Subsequently, it was noted that surgical resection of larger brain metastases might improve the effectiveness of adjuvant radiotherapy and thus contribute to overall survival of these patients [11].

In addition, patients with an increased comorbidity burden were significantly more prone to postoperative complications after surgery for brain metastases. This seems reasonable, since affected patients enter surgery with a much more limited physical constitution. In several other fields of tumor surgery, the Charlson comorbidity index (CCI) has been identified as a valuable tool in identifying patients at risk [12,13]. Considering the influence of comorbidity in a patient on both possible postoperative complications and associated mortality, as described in the present study, the comorbidity burden in a patient should be considered in the benefit–risk analysis of a pending surgical intervention.

With the identification and discussion of the above-mentioned risk factors for postoperative adverse events after surgical resection of BM, the authors do not intend to deprive certain patients with BM of surgical therapy options. Rather, we want to enable a more comprehensive counseling of patients, family members and caregivers based on the awareness of relevant risk factors. In addition, the preoperative identification of patients at risk might facilitate more comprehensive postoperative monitoring and thus contribute to the prevention of some PAEs. Further multicenter-based studies will be needed in order to sufficiently cope with the challenges in the course of interdisciplinary modern treatment and aftercare of patients with brain metastases.

## 4. Materials and Methods

### 4.1. Patients

Between 2013 and 2018, 388 patients suffering from symptomatic brain metastases were surgically treated at the authors’ institution. The study was conducted in accordance with the Declaration of Helsinki, and the protocol was approved by the Ethics Committee of the University Hospital Bonn (No. 250/19). Informed consent was not sought as a retrospective study design was chosen. Patients with leptomeningeal disease were excluded. Patient characteristics, radiological features, laboratory values, location of primary cancer, functional status at admission and during the course of treatment and the presence of early postoperative complications were collected and entered into a computerized database (SPSS, version 25, IBM Corp., Armonk, NY). Furthermore, the comorbidity burden of patients with BM was determined using the Charlson comorbidity index (CCI). The CCI was derived from medical chart reviews and administrative systems [14]. After age adjustment, patients with BM were divided into two groups with CCI < 10 and CCI ≥ 10. The Karnofsky Performance Score (KPS) was used to classify the patients according to their functional status at admission. Patients with BM requiring surgery were evaluated at admission according to their clinical–functional constitution with KPS > 60% or KPS ≤ 60%, as described previously [15]. The KPS stratification cut-off of 70 was chosen according to Péus et al. with regard to the patient’s ability to carry on his or her normal activity and work [16]. In terms of the classification of the American Society of Anesthesiologists (ASA), the patients studied were divided into two groups: preoperative ASA 1 or 2 versus preoperative ASA ≥ 3. The examination of C-reactive protein (CRP) and white blood cells (WBCs) was performed within 12 h after admission as part of the routine laboratory procedures. The WBC counts (normal range 3.9–10.2 G/L) were dichotomized in ≤12 G/L and >12 G/L and the CRP (normal range 0–3 mg/L) in ≤10 mg/L and >10 mg/L. Each brain metastasis was categorized according to the primary site of origin: lung, breast, melanoma, gastrointestinal tract, hematological, prostate, kidney or others. All treatment procedures were determined individually for each patient by consensus during a weekly interdisciplinary tumor board meeting and, if applicable, tailored to the referring physician’s previous therapies.

Concerning the assessment of postoperative complications, further analysis was conducted using a publicly available list of events that the Agency for Healthcare Research and Quality and the Center for Medicare and Medicaid Services refer to as patient safety indicators (PSIs) and hospital-acquired conditions (HACs). PSIs included the complicative occurrence of pressure ulcers, iatrogenic pneumothorax, transfusion reactions, peri- and postoperative hemorrhage, pulmonary embolism, acute postoperative respiratory failure, deep vein thrombosis, postoperative sepsis, wound dehiscence and accidental puncture or laceration. Within the group of HACs, screening was performed for pneumonia, catheter-associated urinary tract infections, fall injuries and vascular catheter-associated infections. In addition, to assess complications specific to cranial surgeries, postoperative periods were screened for iatrogenic postoperative infarction, cerebrospinal fluid (CSF) leakage, postoperative meningitis and ventriculitis, brain edema, cerebrovascular venous thrombosis and postoperative seizures, as well as new or worsened postoperative neurological deficits, and were classified as cranial surgery-related complications (CSCs). Meningitis was defined as a positive microbiological finding in a cerebrospinal culture. Furthermore, meningitis was also defined as a comparable clinical syndrome with predefined cerebrospinal fluid changes in the absence of a confirmatory culture [17,18]. The occurrence of early postoperative complications served as the primary indicator and was defined as any postoperative adverse event (PAE) with or without surgical consequences occurring within 30 days after surgical BM resection.

### 4.2. Statistics

Data analyses were performed using the computer software package SPSS (version 25, IBM Corp., Armonk, NY). Categorical variables were analyzed in contingency tables using Fisher’s exact test. Results with *p* < 0.05 were considered statistically significant. Area under the curve (AUC), specificity and sensitivity of the age-adjusted CCI for predicting postoperative complications in the present patient population were determined using the ROC curve. OS was analyzed by the Kaplan–Meier method using the Gehan–Breslow–Wilcoxon test. Results with *p* < 0.05 were considered statistically significant.

## 5. Limitations

The present study has several limitations. Acquisition of data was retrospective. Additionally, the present data represent a single-center experience only. Potential selection bias can be presumed since due to the nature of the question only patients with surgical treatment were included. Nevertheless, data are presented which provide an unprecedented insight into the perioperative settings of neurosurgical resection of brain metastases. Furthermore, these data represent interdisciplinary treatment decisions.

## 6. Conclusions

Information gained about the risk factors for postoperative complications enables future prevention and optimal response, especially in at-risk oncological patients. The present study identified the presence of multiple brain metastases and an increased comorbidity burden as risk factors for postoperative adverse events—whether PSIs, HACs or specific CSCs—in patients suffering from brain metastases.

## Figures and Tables

**Figure 1 cancers-12-03209-f001:**
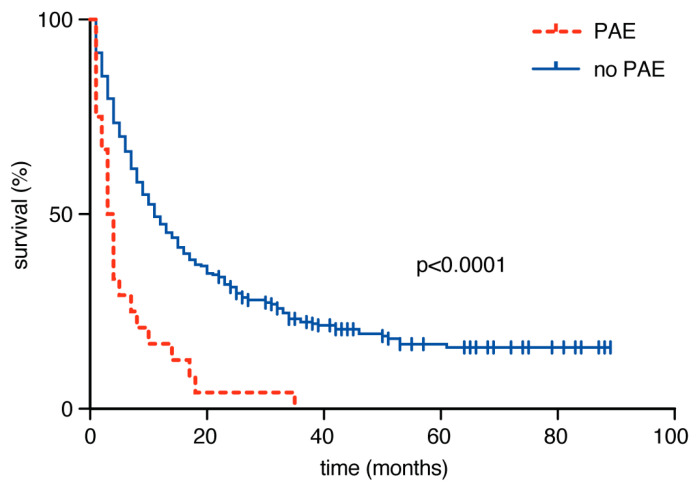
Kaplan–Meier curves for overall survival (OS) dependent on the occurrence of postoperative adverse events (PAEs).

**Table 1 cancers-12-03209-t001:** Patient characteristics. Values are presented as the number of patients (%) unless stated otherwise. ASA, American Society of Anesthesiology; CRP, C-reactive protein; KPS, Karnofsky Performance Score; n.s., not significant; SD, standard deviation; WBC, white blood cell.

Patient Characteristics	Patients without PAE 355 (91)	Patients with PAE 33 (9)	*p*-Value
Mean age (±SD) (years)	63 ± 12	66 ± 9	n.s.
Female sex	172 (49)	19 (58)	n.s.
Mean LOS (±SD) (days)	14 ± 12	26 ± 22	<0.0001
Primary site of cancer			
Lung	152 (43)	13 (39)	n.s.
Breast	48 (14)	2 (6)	n.s.
Melanoma	36 (10)	5 (15)	n.s.
Gastrointestinal	49 (14)	3 (9)	n.s.
Other	70 (20)	10 (30)	n.s.
Multiple intracranial lesions	108 (30)	17 (52)	0.02
Preoperative KPS ≥ 70	310 (87)	26 (79)	n.s.
ASA ≥ 3	193 (54)	21 (64)	n.s.
Preoperative CRP > 10 mg/L	78 (22)	12 (36)	n.s.
Mean preoperative CRP (mg/L)	9.2 ± 17.9	16.7 ± 35.1	0.04
Preoperative WBC count > 12 G/L	168 (47)	13 (39)	n.s.
Mean preoperative WBC count (G/L)	12.7 ± 5.9	12.1 ± 5.3	n.s.
Preoperative intake of coagulation-promoting medication	70 (20)	5 (15)	n.s.
CCI ≥ 10	174 (49)	25 (76)	0.003
In-hospital deaths	2 (0.6)	8 (24)	<0.0001

**Table 2 cancers-12-03209-t002:** Overview of postoperative adverse events. Values represent number of patients (%) unless otherwise indicated. CSCs, cranial surgery-related complications; CSF, cerebrospinal fluid; HACs, hospital-acquired conditions; PAEs, postoperative adverse events; PSIs, patient safety indicators.

Postoperative Adverse Events	N
No. of patients	388 (100)
No. of PAE	33 (8.5)
**PSIs**	17 (4.4)
Postoperative hemorrhage	11 (2.8)
Postoperative respiratory failure	3 (0.8)
Perioperative pulmonary embolism	2 (0.5)
Wound dehiscence	1 (0.3)
**HACs**	5 (1.3)
Pneumonia	3 (0.8)
Catheter-associated urinary tract infection	2 (0.5)
**Specific CSCs**	11 (2.8)
Brain edema	3 (0.8)
Meningitis	3 (0.8)
CSF leakage	2 (0.5)
Postoperative seizures	2 (0.5)
Cerebrovascular venous thrombosis	1 (0.3)

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
