# Peer review of "Comorbidity Burden and Presence of Multiple Intracranial Lesions Are Associated with Adverse Events after Surgical Treatment of Patients with Brain Metastases"

_cancers, 2020, doi:10.3390/cancers12113209_

Round 1

Reviewer 1 Report

This is a well written and important paper describing predictors of post operative morbidity after resection brain metastases.  This is a very important clinical paper as many patients who are at higher risk of operative morbidity can be safely treated with radiosurgery.  Overall the paper is very clear and concise and has sound conclusions.  My only question is related to the statisitics, as I am not as familier with the technique of using an ROC curve to establish a predicctive model.  Is there a reason that multivariable linear regression was not utilized to determine the odd ratio of toxicity.  Also which variables were used for the model?  Was it all the factors in Table 1.

Author Response

Dear reviewer,

thank you for your interest in our work as well as the fair and constructive assessment of our manuscript cancers-963661 (Comorbidity burden and presence of multiple intracranial lesions are associated with adverse events after surgical treatment of patients with
brain metastases). As far as possible, we have addressed all raised concerns and taken up all the reviewer’s suggestions. In order to markedly improve the value of the manuscript, indicated changes were made to the manuscript, as outlined below.

Comment 1:

„My only question is related to the statisitics, as I am not as familier with the technique of using an ROC curve to establish a predicctive model.  Is there a reason that multivariable linear regression was not utilized to determine the odd ratio of toxicity. Also which variables were used for the model?  Was it all the factors in Table 1.“

Authors‘ response:

Thank you for this comment on the statistical analysis. The reason to decide for an ROC is outlined in the following. The aim of our analysis was to find a simple predictive model for PAE based on the comorbidity burden, assessed by the CCI. The ROC-analysis is a tool to evaluate predictive performance of a single continuous variable/marker (Zou et al., 2007; PMID: 17283280), e.g. by determining the AUC to summarize the diagnostic accuracy. Additionally, based on the ROC we were able to determine the best-performing cut-off point (the CCI ranges from 0 to 12) in terms of sensitivity and specificity. After intensive statistical counselling, we decided to add Dr. Weinhold as an additional coauthor.

Again, we would like to thank all the reviewers for thoroughly reading our manuscript as well as for the comments to further improve the manuscript.

We believe that after incorporating the issues listed above, the manuscript is much clearer for potential readers.

We hope that the manuscript is now eligible for publication in Cancers.

Sincerely yours

Matthias Schneider

(on behalf of the authors)

Reviewer 2 Report

In the present study, the authors investigated potential patient-specific risk factors for occurrence of postoperative complication in surgical treated brain metastases. The reviewed manuscript focuses on a topic of highest interest. For although this is a subject area that has already been sufficiently addressed, the focus on postoperative complications and their prevention is of enormous importance in this seriously ill patient group. Despite the well-written manuscript, I have listed below some minor suggestions to help the authors to further improve their manuscript prior to acceptance:

Methods:

- Minor: Please insert IRB approval number.

- Minor: The authors should insert appropriate references for the selected cut-of-values of the dichotomization of included variables or present the corresponding calculation.

- Minor: Please explain the definition used for the postoperative complication of "meningitis". Only based on microbiological findings? Based on MRI scans?

Results:

- Major: Despite the focus on the occurrence of postoperative complications, the authors should also provide date on their impact on survival. For example, OS or at least the 1-y mortality (due to the multiple aetiologies)

- Major: They should then also present a corresponding figure with a graphical representation of these data.

Table 1

- Minor: The percentage distribution of patient groups does not add up to 100.

Discussion:

- Minor: The discussion should conclude by mentioning potential future research initiatives. Where do the authors see further academic interest based on their results? Prospective studies? Multi-center projects? Register studies?

Limitations:

- Minor: Due to the clear limitations of the present study, this section should be extended. Selection bias? Excluded patients?

By incorporating the listed suggestions for improvement prior to acceptance of the manuscript, I think that this manuscript - due to the novel view on a current, approved clinical problem - is worth a publication in the Special Issue of Cancers.

Author Response

Dear reviewer,

thank you for your interest in our work as well as the fair and constructive assessment of our manuscript cancers-963661 (Comorbidity burden and presence of multiple intracranial lesions are associated with adverse events after surgical treatment of patients with
brain metastases). As far as possible, we have addressed all raised concerns and taken up all the reviewer’s suggestions. In order to markedly improve the value of the manuscript, indicated changes were made to the manuscript, as outlined below.

Comment 1:

“Minor: Please insert IRB approval number.”

Authors’ response/Changes to the manuscript:

IRB approval number was inserted.

Comment 2:

“The authors should insert appropriate references for the selected cut-of-values of the dichotomization of included variables or present the corresponding calculation.”

Authors’ response:

Due to the fact that metastatic cancer disease represents a variable within the CCI the established cut-off ≥2 does not fit for the present analysis. We therefore decided to determine a cut-off adjusted to our setting of selected brain metastatic patients. The ROC-analysis is a tool to evaluate predictive performance of a single continuous variable/marker (Zou et al., 2007; PMID: 17283280), e.g. by determining the AUC to summarize the diagnostic accuracy. Additionally, based on the ROC we were able to determine the best-performing cut-off point (the CCI ranges from 0 to 12) in terms of sensitivity and specificity. After intensive statistical counselling, we decided to add Dr. Weinhold as an additional coauthor.

Comment 3:

„Please explain the definition used for the postoperative complication of "meningitis". Only based on microbiological findings? Based on MRI scans?“

Authors‘ response:

Meningitis was defined as a positive microbiological finding in cerebrospinal fluid culture. Furthermore, meningitis was also defined as a comparable clinical syndrome with predefined cerebrospinal fluid changes in the absence of a confirmatory culture (PMID: 15857548, 28938497).

Changes to the manuscript:

The following addition was made within the methodological section:

„Meningitis was defined as a positive microbiological finding in cerebrospinal culture. Furthermore, meningitis was also defined as a comparable clinical syndrome with predefined cerebrospinal fluid changes in the absence of a confirmatory culture. (PMID: 15857548, 28938497).“

Comment 4:

“Major: Despite the focus on the occurrence of postoperative complications, the authors should also provide date on their impact on survival. For example, OS or at least the 1-y mortality (due to the multiple aetiologies). They should then also present a corresponding figure with a graphical representation of these data.”

Authors’ response:

Following the reviewer’s suggestions, we decided to additionally analyze the presented cohort of patients with cranial metastases with regard to a potential correlation of OS and PEA. Patients without PEA exhibited a mOS rate of 11 months (95% CI 9-13) compared to 2.5 months (95% CI 1-4 months) for patients with PEA (p<0.0001).

Changes to the manuscript:

The results section was expanded by the following:

“Analysis of OS revealed that patients without PEA exhibited a mOS rate of 11 months (95% CI 9-13) compared to 2.5 months (95% CI 1-4 months) for patients with PEA (p<0.0001). “

Furthermore, Figure 1 was incorporated into the manuscript and depicts a Kaplan Meier analysis of mOS dependent on the presence of PEA. The following section was implemented into the Methods:

“OS was analyzed by the Kaplan–Meier method using Gehan–Breslow–Wilcoxon test. Results with p < 0.05 were considered statistically significant.”

Comment 5:

„Minor: The percentage distribution of patient groups does not add up to 100.“

Authors‘ response/Changes to the manuscript:

Respective values were corrected.

Comment 6:

„Minor: The discussion should conclude by mentioning potential future research initiatives. Where do the authors see further academic interest based on their results? Prospective studies? Multi-center projects? Register studies?“

Authors‘ response/Changes to the mansucript:

Following the reviewer’s suggestions, the final part of the discussion section now reads as follows:

„With the identification and discussion of the above-mentioned risk factors for postoperative adverse events after surgical resection of BM, the authors do not intend to deprive certain patients with BM of surgical therapy options. Rather, it is our concern to enable a more comprehensive counseling of patients, family members and caregivers based on the awareness of relevant risk factors. In addition, the preoperative identification of patients at risk might facilitate a more comprehensive postoperative monitoring and thus contribute to the prevention of some PAE. Further multicenter-based studies will be needed in order to sufficiently cope with the challenges in the course of interdisciplinary modern treatment and aftercare of patients with brain metastases.“

Comment 7:

“Minor: Due to the clear limitations of the present study, this section should be extended. Selection bias? Excluded patients.“

Authors‘ response/Changes to the manuscript:

We decided to expand the limitations section, which now reads as follows:

„The present study has several limitations. Acquisition of data was retrospective. Additionally, the present data represent a single-center experience, only. Potential selection bias can be presumed since due to the nature of the question only patients with surgical treatment were included. Nevertheless, data are presented which provide an unprecedented insight into the perioperative settings of neurosurgical resection of brain metastases. Furthermore, this data represents interdisciplinary treatment decision.“

Again, we would like to thank all the reviewers for thoroughly reading our manuscript as well as for the comments to further improve the manuscript.

We believe that after incorporating the issues listed above, the manuscript is much clearer for potential readers.

We hope that the manuscript is now eligible for publication in Cancers.

Sincerely yours

Matthias Schneider

(on behalf of the authors)